# Impact of vitamin C on the reduction of opioid consumption for acute musculoskeletal pain: A double-blind randomized control pilot study

**Raoul Daoust**[1,2,3]*, **Jean Paquet**[1], **David Williamson**[3,4], **Vérilibe Huard**[1,2], **Caroline Arbour**[3,5], **Jeffrey J. Perry**[6,7], **Marcel Émond**[8], **Simon Berthelot**[8,9], **Patrick Archambault**[8,10], **Dominique Rouleau**[3], **Judy Morris**[1,2,3], **Alexis Cournoyer**[1,2,3]

1 Study Center in Emergency Medicine, Hôpital du Sacré-Coeur de Montréal (CIUSSS du Nord-de-l'Île de-Montréal), Montréal, Québec, Canada, 2 Département de Médecine Familiale et de Médecine d'Urgence, Faculté de Médecine, Université de Montréal, Montréal, Québec, Canada, 3 Centre de Recherche de l'Hôpital du Sacré-Coeur de Montréal (CIUSSS du Nord de-l'Île-de-Montréal), Montréal, Québec, Canada, 4 Faculté de Pharmacie, Université de Montréal, Montréal, Québec, Canada, 5 Faculté des Sciences Infirmières, Université de Montréal, Montréal, Québec, Canada, 6 Department of Emergency Medicine, University of Ottawa, Ontario, Canada, 7 Ottawa Hospital Epidemiology Program, Ottawa Hospital Research Institute, Ontario, Canada, 8 Département de Médecine de Famille et de Médecine D'urgence, Faculté de Médecine, Université Laval, Québec, Québec, Canada, 9 Axe Santé des Populations et Pratiques Optimales en Santé, Centre de recherche du CHU de Québec-Université Laval, Québec, Québec, Canada, 10 Centre de Recherche Intégré pour un Système Apprenant en Santé et Services Sociaux, Centre Intégré de Santé et de Services Sociaux de Chaudière-Appalaches, Lévis, Québec, Canada

* raoul.daoust@umontreal.ca

## Abstract

### Introduction

Recent evidence has shown that vitamin C has analgesic and opioid sparing properties in immediate postoperative context. However, this has never been studied for acute musculo-skeletal (MSK) emergency department (ED) injuries. The aim of this pilot study is to evaluate the feasibility of conducting a randomized placebo-controlled study to determine the opioid sparing and analgesic effect of vitamin C compared to placebo, in acute MSK injured ED patients.

### Methods

A double-blind randomized controlled trial (RCT) distributed in two arms, stratified for fractures, was performed in a tertiary care center, one group receiving 1 g of vitamin C twice a day for 14 days and another receiving placebo. Participants were ≥18 years of age, treated in ED for MSK injuries present for ≤2 weeks, and discharged with a standardized opioid pre-scription of 20 morphine 5 mg tablets (M5T) and, at the clinician discretion, 28 tablets of naproxen 500 mg. Participants completed a 14-day paper diary and were contacted by phone at 14 days, to document their analgesic use, vitamin C consumption, and pain intensity.

**Data Availability Statement:** Our ethical committee does not permit that data from patients can be made publicly available for ethical reason because it would compromise patient privacy. Data can be requested from authors who will obtain permission from the ethical committee. E-mail: C omite.Ethique.Recherche.cnmtl@ssss.gouv.qc.ca.

**Funding:** This research was supported by the « Fonds Alma Mater et Chaire Docteur Sadok Besrour de l'Université de Montréal », the « Association des spécialistes en médecine d'urgence du Québec » and the « Fonds des Urgentistes de l'Hôpital du Sacré-Cœur de Montréal ». The funding sources had no role in the design and conduct of the study; collection, management, analysis, and interpretation of the data; preparation, review, or approval of the manuscript; and the decision to submit the manuscript for publication.

**Competing interests:** The authors have declared that no competing interests exist.

## Results

Overall, 137 patients were screened; 44(32%) were excluded, 38(40.9%) refused, leaving 55(59.1%) participants, with a consent rate of 9.2/month. Mean age was 53 years (SD = 16) and 55% were men. Fourteen (25%) participants were lost to follow-up and 33(83%) patients complied with treatment. For per-protocol analysis, the median (IQR) M5T consumed was 6.5 (3.3–19.5) for the vitamin C and 9.0 (1.5–16.0) for placebo group. The median (IQR) naproxen 500 mg tablets consumed was 0 (0–9.8) for the vitamin C group and 20 (0–27) for the placebo arm.

## Conclusion

This pilot study supports the feasibility of a larger RCT on the opioid sparing and analgesic properties of vitamin C for acute MSK injured ED patients. Strategies to reduce the refusal and lost to follow-up rates are discussed.

## Trial registration number

NCT05555576, ClinicalTrials.Gov PRS.

## Introduction

Opioids remain an important part of the treatment for moderate to severe acute pain in the emergency department (ED) and are frequently prescribed for home pain management after ED discharge [1, 2]. However, even short-term opioids use after an ED visit can cause undesirable side effects like constipation, nausea/vomiting, drowsiness, weakness [3], and more severe adverse events like respiratory depression [4]. It can also lead to long-term use [5, 6], opioid use disorders [7], overdose, and death [8, 9]. Furthermore, larger quantity of opioids consumed is associated with a higher prevalence of opioid-related adverse events [10]. Therefore, reducing opioid consumption for acute pain without compromising pain management is a constant challenge for clinicians [11].

Currently, the main strategy employed to reduce opioid consumption is to limit the rate and quantity of opioids prescribed for acute pain [12–15]. However, most interventions reduce opioid prescription rates, but not the quantity of prescribed opioids [16]. One strategy to reduce opioid consumption is the use of adjunct analgesia, and non-steroid anti-inflammatory drugs (NSAIDs) are frequently suggested for this purpose. However, NSAIDs are associated with significant gastrointestinal, cardiovascular, musculoskeletal, and renal adverse effects [17–19]. Recent evidence has shown that vitamin C (ascorbic acid), possesses analgesic effects as well as antioxidant properties [20–22]. One potential mechanism involves reducing the production of free radicals, which helps protect tissues (including nerves) from irreversible damage [23, 24]. Furthermore, vitamin C is associated with very low toxicity and rare adverse effects [25].

Two systematic reviews reported the effects of vitamin C on opioid consumption for immediate postoperative pain. One showed significant reductions in opioid requirements and pain scores up to 24 hours postoperative with intravenous vitamin C [26]. The other found a moderate-level evidence supporting the use of 2g of oral vitamin C preoperative to reduce postoperative morphine consumption [27]. One of the included studies administered 500 mg of vitamin C intravenously twice a day up to the third day post-surgery, showing reduced pain

intensity and opioid consumption compared to controls [28]. Furthermore, in two recent small studies, administering 200 mg of vitamin C orally three times a day for 10 days or 500 mg orally twice daily for seven days following tooth extractions was associated with reduced postoperative pain up to 72 hours after surgery [29, 30].

### Study rationale

The literature suggests that administering vitamin C can reduce both pain and opioid consumption in the context of immediate postoperative acute pain (24 to 72 hours) [26–30]. The mechanisms underlying postoperative pain are analogous to those responsible for pain in musculoskeletal trauma observed in the ED [31, 32]. To our knowledge, there is no evidence addressing the effectiveness of vitamin C administration following an ED visit for acute pain, particularly in MSK trauma injuries such as fractures, contusions, sprains, and strains. Furthermore, previous studies have only assessed opioid consumption and pain relief during the immediate postoperative period of 24 to 72 hours. Since the need for analgesics, including opioids, may persist for up to two weeks in some ED-discharged patients which is in line with the usual acute pain definition [33, 34].

### Study objectives

The aim of this pilot study was to assess the feasibility of conducting a randomized, placebo-controlled trial to determine the opioid-sparing and analgesic effects of vitamin C compared to placebo (lactose) over a two-week follow-up period in ED-discharged patients with acute musculoskeletal (MSK) injuries. The secondary aims were to obtain preliminary results on opioid and other analgesic consumption, and average pain intensity during a two-week follow-up for patients receiving vitamin C and those receiving a placebo.

## Methods and analyses

### Study design

We conducted a double-blind, randomized, placebo-controlled pilot study in one tertiary trauma care university-affiliated hospitals located in Montreal (Québec, Canada) with an annual census of 60,000 visits. This parallel group pilot randomized trial is reported in accordance with the CONSORT 2010 (Consolidated Standards of Reporting Trials) statement for reporting parallel group randomized trial [35]. This pilot study is based on a recently published protocol [36].

### Patient and public involvement

A patient partner was involved in the design of the study and the pain medication diary used in this study. This study was approved by the local Ethics Review Committee.

### Participants and recruitment

Consecutive patients (from 8 to 16h on weekdays) diagnosed with acute MSK injury pain complaint ongoing for less than two weeks and discharged from the ED with an opioid prescription were approached by the treating clinician to participate in the study and obtain their verbal consent to be seen by a research assistant. Patients were recruited from March 24th 2023 to October 31 2023. The decision to prescribe opioids was at the treating physician's discretion. The research assistant then verified the patient's inclusion and exclusion criteria, explained the research protocol, and obtained informed written consent.

**Eligibility criteria.**   Patients were eligible for inclusion if they satisfied all following criteria: (1) aged 18 and over; (2) treated in ED for acute MSK injury pain present for less than two weeks; (3) discharged with an opioid prescription; and (4) French or English-speaking.

Patients were excluded from the study if any of the following criteria applied: (1) opioid use in the month prior to ED visit; (2) already taking vitamin C supplement [37]; (3) active cancer; (4) treated for chronic pain; (5) treated for opioid use disorder; (6) unable to fill out a diary or unavailable for follow-up; (7) any allergy to milk or morphine; (8) treated with cyclosporine or warfarin; (9) Pre-existing oxalate nephropathy or hemochromatosis [38, 39]; or (10) pregnant or lactating (dosage vitamin C > 1,800 mg is not recommended) [40] for women of childbearing age a urine pregnancy test was performed.

**Randomization method and blinding.**   Eligible patients were block randomized (variable size) at the initial visit (via 1:1 ratio) to either 1 000 mg vitamin C taken orally twice a day or placebo (lactose) (Fig 1), using a centralized randomization web system. Since fractures are associated with more opioid consumption, randomization was stratified by presence of a fracture (Yes or No) [34, 41]. In accordance with the centralized web system, an independent blinded pharmacist dispensed pre-packed numbered bottles of identical capsule of either vitamin C or placebo capsules.

## Study drug

Vitamin C is a vital nutrient; it helps form and maintain bones, skin, and blood vessels, and has antioxidant properties. It is not produced by the human body, but occurs naturally in fruits, vegetables, and other nutriments. It is also available as a supplement over the counter in pharmacies, supermarkets, health stores, and online. Considering dosages used in studies performed in acute postoperative and chronic pain contexts, adverse events reported with different dosages, and the absorption of vitamin C given orally [36], we compared the effect of 1,000 mg of vitamin C taken orally twice a day (morning and evening) for a 14-day period after ED discharge for the treatment arm to a placebo (lactose) for the control arm. Based on our prior research to determine the appropriate quantity of opioids for a two-week supply for most patients [34], all participants were also provided with a standardized prescription for 20 tablets of 5 mg morphine. Clinicians could also prescribe 28 tablets of naproxen 500 mg (NSAID) at their discretion.

**Subject compliance monitoring.**   All participants were asked to complete a 14-day paper diary. Each day, participants were asked if they took their study capsule and if they started using a new vitamin or natural products. Phone calls were used as a reminder to fully complete and return the diary. Results from our previous work [3, 34, 42] suggest that nearly all patients fully understand how to fill out our diary and questionnaire.

## Study procedures

After verifying eligibility criteria, research assistants documented the following initial data in REDCap (Vanderbilt University, TN) electronic data capture tools hosted at Hôpital Sacré-Coeur de Montréal [43]: unique patient identifier, demographic variables (sex, gender, age, and ethnicity), phone numbers, ED length of stay, final diagnosis (injury type and severity), history of opioid use, opioids received during ED stay, pain intensity at triage and discharge, pain medication prescribed at discharge, vitamins and natural products consumed, if it was a work-related injury and if they have salary insurance coverage.

A pharmacist dispensed either active or placebo capsules and patients were informed not to start consuming any other vitamin or natural health products. All participating patients were given the validated 14-day paper diary, and the research assistant completed the baseline day 1

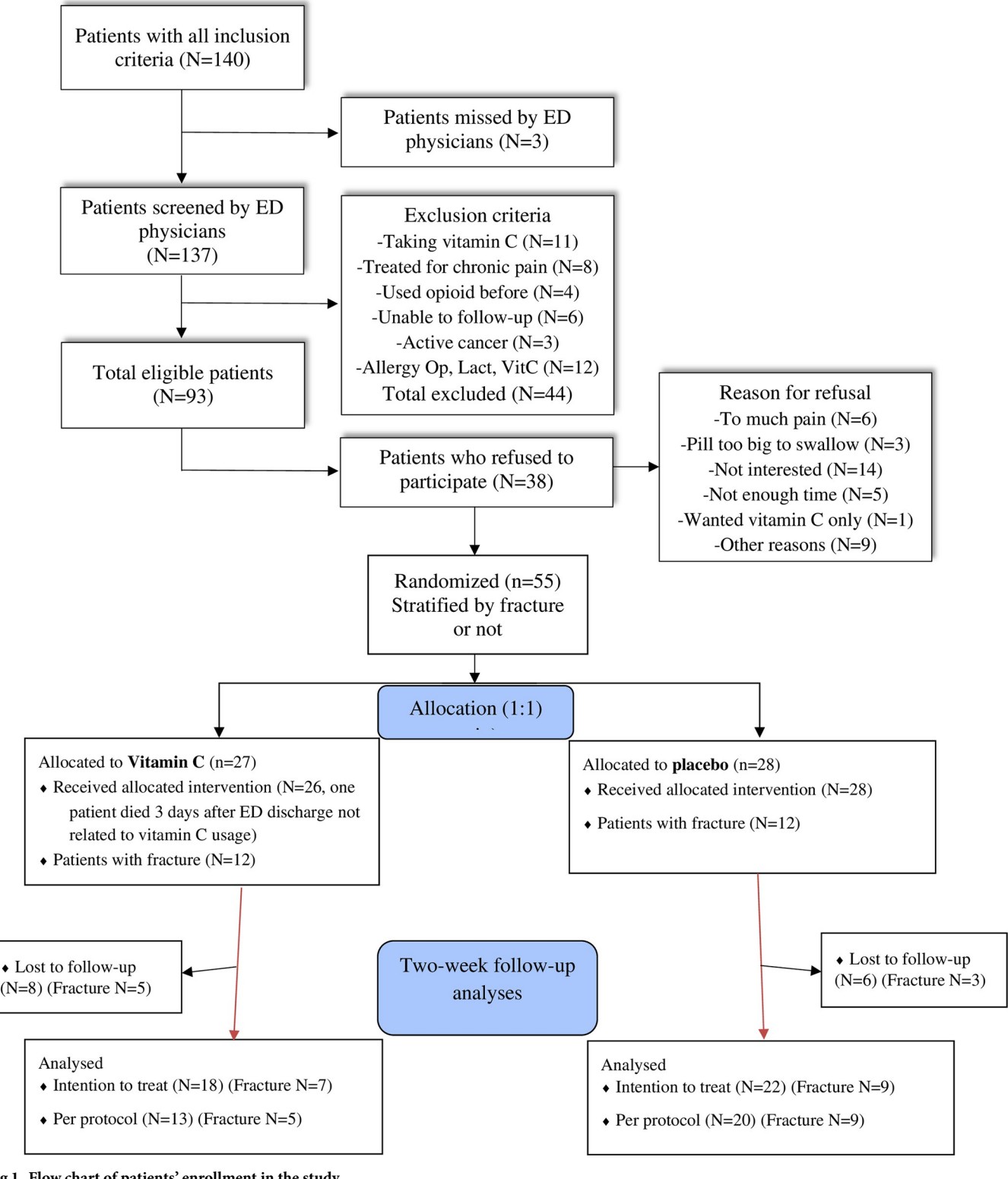

**Fig 1. Flow chart of patients' enrollment in the study.**

questionnaire. Patients were instructed in person on how to use the paper diary and to treat their pain, starting with acetaminophen and NSAIDs (if prescribed) before consuming opioids. Patients also had phone access to research assistants should they require assistance.

The diary was used daily for real-time recording of quantity, date, and names of all pain medications consumed, including vitamin C, related to the patient's ED visit. At the end of each day, pain intensity was assessed by one question: "What was your pain level throughout the day?" measured on an 11-point numerical rating scale (NRS) from 0 to 10 (0 represents no pain and 10 the worst pain imaginable) and in general, if their pain had been relieved during the day (Yes or No).

At the end of two weeks, patients were contacted by phone. A series of questions were asked regarding their actual pain intensity level as measured using the same 11-point NRS, if they filled their initial opioid prescription, if they received additional opioid prescriptions, their reasons for stopping or not taking any opioids (i.e., not enough or no pain, overly severe side effects, fear of addiction, or other), clinician visits related to their initial pain condition, if their medications were adequate for their overall pain management, and if they had surgery related to their initial pain condition. They were asked if they used any other product to relieve pain (e.g., cannabis, alcohol, vitamins, other natural or over-the-counter products, etc.). As mitigation strategy, we counted the remaining morphine tablets and study capsules which allowed us to retrieve our outcome for participants without (or incomplete) 14-day diaries. It was planned that in the event of an unexpected adverse reaction that was fatal or life-threatening, the research team would notify the Natural Health Products Directorate as soon as possible, within seven days of becoming aware of it. If the unexpected serious adverse reaction was not fatal or life-threatening, notification would be made no later than 15 days after the research team became aware of it.

## Outcomes

The main outcome for feasibility was the recruitment rate. Secondary outcomes were: missed rates, exclusion rates, consent rates, follow-up rates, reasons for exclusion and refusal, and treatment compliance rates. Compliance to treatment was defined as taking vitamin C or placebo capsules at least 13 days out of 14. To assess feasibility, our results were compared to those reported in a recent review of randomized controlled trials; 72% (IQR: 50%–88%) median consent rate and 88% (IQR: 80%–97%) median retention rate [44]. A previous study on a similar population (63% MSK), but with a 24/7 recruitment time frame, recruited 60 patients per month [34]. Since we were recruiting only during weekdays and only MSK injured patients, we expected a maximum of nine patients per month to be eligible for this study. We therefore defined feasibility as at least 6.5 participants per months, using the median consent rate reported in randomized controlled trials (72%) [44]. We will also report quantity of analgesics consumed, including morphine 5 mg tablets, and pain intensity during the two-week follow-up.

## Sample size

According to guidelines for designing pilot studies, ideally 30 participants should be recruited per group [45]. In a previous study with similar outcomes, we observed a lost to follow-up rate of 18%, so we planned to recruit 36 participants per group [34].

## Statistical analysis

We performed an intention-to-treat analysis and per-protocol analysis (participants with perfect compliance and follow-up). Since this study was not powered as a superiority randomized

controlled trial (RCT), the per-protocol analysis was included to identify potential treatment effects because non-compliance and crossover tend to attenuate the between-arm differences [46]. We used descriptive statistics to compare baseline characteristics between included participants, those who refused to participate, and participants lost to follow-up, using means with their standard deviation (SD) for continuous variables, median with interquartile range (IQR) for non-normal distributed variables, and proportions for categorical variables. The same descriptive statistics were used to examine socio-demographic variables and analgesic prescriptions at discharge for participants in each group. Median number (IQR) morphine 5mg tablets and other analgesic consumed during the two-week follow-up were presented for both vitamin C and placebo arms. Finally, mean pain intensity with standard error of the mean (SEM) for each day of follow-up were illustrated for both vitamin C and placebo groups.

## Results

### Study cohort description

Fifty-five participants were randomized with a mean age (SD) of 52 (17) years, 54% were male, and mean (SD) pain intensity at triage was 8.0 (1.5). Included participants were relatively similar for most baseline characteristics compared to those who refused to participate (Table 1). However, more female refused to participate (60.5%) than study participants (46.3%). Also, more participants were treated with an opioid during their ED stay compared to patients that refused to participate (59.3% vs 36.8%). Participants' ED stay also tended to be shorter ED compared to patients who refused to participate (6.4 vs 8.2 hours).

Socio-demographic variables and analgesic prescriptions at discharge for participants in the vitamin C and placebo group are presented in Table 2. The vitamin C group was composed of only 28% of women compared to 59% for the placebo group. All other socio-demographic variables were similar between both groups. Almost all vitamin C participants received an acetaminophen prescription at discharge compared to 73% for the placebo participants. Half the vitamin C participants had naproxen (NSAID) prescription at discharge compared to 73% of the placebo participants. Also, 16.6% of the vitamin C group had an additional opioid prescription and 16.6% had a surgical intervention during follow-up compared to 5% and 0% for the placebo group, respectively.

**Table 1. Comparison of baseline characteristics between patients who refused to participate and those who accepted to participate in the study.**

| Baseline characteristics | Refused to participate (N = 38) | Participants (N = 55) |
|---|---|---|
| Mean (±SD) age | 48.2 (19.9) | 51.8 (17.2) |
| Female (%) | 60.5 | 46.3 |
| Mean (±SD) pain intensity (0–10 scale) at triage | 7.3 (1.7) | 8.0 (1.5) |
| Type of pain conditions (%) | | |
| Fracture | 31.6 | 40.7 |
| Contusion | 21.1 | 9.3 |
| Back pain | 28.9 | 25.9 |
| Neck pain | 5.3 | 11.1 |
| Other musculoskeletal pain | 13.2 | 13.0 |
| Treated with opioid within the ED stay (%) | 36.8 | 59.3 |
| Median (Q1-Q3) ED stay (hours) | 8.2 (4.3–11.2) | 6.4 (3.9–9.0) |

Q1-Q3: first and third quartiles; ED: emergency department; SD: standard deviation

**Table 2. Baseline characteristics and analgesic prescription between vitamin C and placebo arms for participant with follow-up.**

| Baseline characteristics and analgesic prescription | Vitamin C (N = 18) | Placebo (N = 22) |
|---|---|---|
| Mean (±SD) age | 54.9 (17.1) | 51.0 (14.9) |
| Female (%) | 27.8 | 59.1 |
| Ethnicity (%) | | |
| • White | 55.6 | 59.1 |
| • Other | 44.4 | 40.9 |
| Education level (%) | | |
| • Primary/High school | 22.3 | 18.2 |
| • College/University | 77.7 | 81.8 |
| Smoker (%) | 16.7 | 13.6 |
| Used opioid in the past year (%) | 5.6 | 9.1 |
| Mean (±SD) pain intensity (0–10 scale) at triage | 7.7 (1.7) | 8.5 (1.1) |
| Mean (±SD) pain intensity (0–10 scale) at ED discharge | 5.7 (2.9) | 6.6 (2.0) |
| Type of pain conditions (%) | | |
| Fracture | 38.9 | 40.9 |
| Contusion | 5.6 | 9.1 |
| Back pain | 27.8 | 36.4 |
| Neck pain | 11.1 | 4.5 |
| Other musculoskeletal pain | 16.7 | 9.1 |
| Treated with opioid within the ED stay (%) | 55.6 | 50.0 |
| Received acetaminophen prescription at discharged (%) | 94.4 | 72.7 |
| Received NSAIDs prescription at discharged (%) | 50.0 | 72.7 |
| Received additional opioid prescription during follow-up (%) | 16.6 | 4.5 |
| Had surgery during the two weeks follow-up (%) | 16.6 | 0 |

Q1-Q3: first and third quartiles; ED: emergency department; SD: standard deviation

## Feasibility outcomes

During the recruitment period, 137 patients were screened (3 patients were missed by physicians). Of these, 44 (32%) patients were excluded (Fig 1). Main reasons for exclusion were allergies to opioids or lactose, already taking vitamin C supplements, and suffering from chronic pain. Of the 93 eligible patients, 38 (41%) declined to participate, mainly because they were not interested in the study, were in too much pain or did not have enough time to participate. Consequently, 55 participants were randomized, 27 were assigned to the vitamin C group and 28 to the placebo group (12 participants in each group had fractures). The recruitment rate was 9.2 participants per month. A total of 14 (25%) participants were lost to follow-up at two weeks, their baseline characteristics were similar to those who completed the follow-up, except that they were treated with opioids more often during their ED stay (Table 3). Compliance with treatment was 83% and it was greater in the placebo group (20/22) compared to the vitamin C group (13/18). Baseline characteristics were similar between compliant and non-compliant participants (see S1 Table) and non-compliance was associated with less pain intensity in the second week of follow-up (Fig 2).

## Opioid and analgesic consumption during follow-up

Analgesic consumption during the 14-day follow-up between participants receiving vitamin C and those receiving placebo for the intention-to-treat and per-protocol analyses are reported in Table 4. For the intention to treat analysis, the median number of morphine 5mg tablets

**Table 3. Comparison of baseline characteristics between participants who were lost to follow-up and those who completed follow-up.**

| Baseline characteristics | Lost to follow-up (N = 14) | Completed follow-up (N = 40) |
|---|---|---|
| Mean (±SD) age | 49.2 (21.1) | 52.7 (15.8) |
| Female (%) | 50.0 | 45.0 |
| Mean (±SD) pain intensity (0–10 scale) at triage | 7.5 (1.8) | 8.1 (1.4) |
| Type of pain conditions (%) | | |
| Fracture | 42.9 | 40.0 |
| Contusion | 14.3 | 7.5 |
| Back pain | 7.1 | 32.5 |
| Neck pain | 21.4 | 7.5 |
| Other musculoskeletal pain | 14.3 | 12.5 |
| Treated with opioid within the ED stay (%) | 78.6 | 52.5 |
| Median (Q1-Q3) ED stay (hours) | 7.3 (4.9–11.2) | 5.3 (3.7–8.6) |

Q1-Q3: first and third quartiles; ED: emergency department; SD: standard deviation

consumed was higher in the vitamin C group than the placebo group (8.5 vs 6.0 tablets). However, the median number of naproxen tablets consumed was substantially lower in the vitamin C group (0 vs 15 tablets). When we examine the per-protocol analysis, the median number of morphine 5mg tablets consumed was lower in the vitamin C group than the placebo group (6.5 vs 9.0 tablets). Again, the naproxen consumption was substantially lower in the vitamin C group (0 vs 20 tablets).

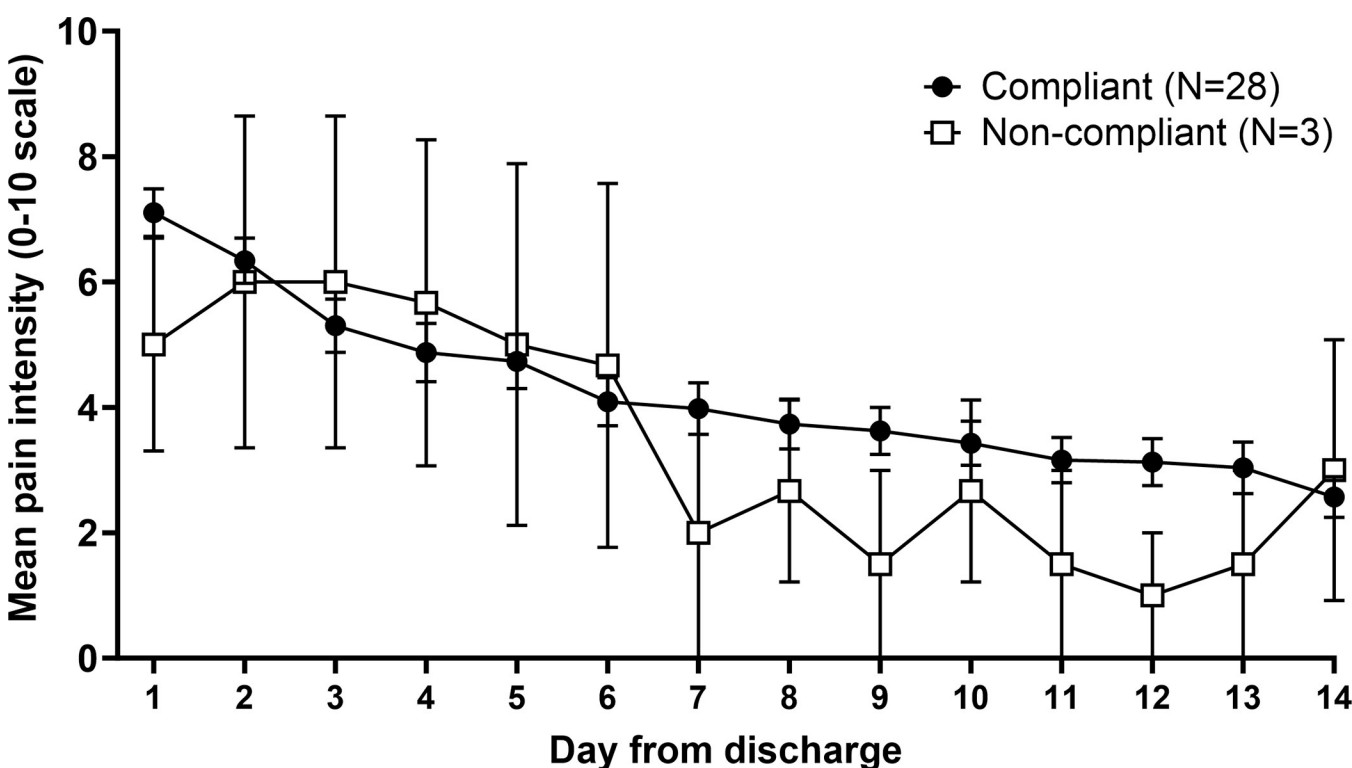

**Fig 2. Mean (SEM) daily pain intensity (0–10 scale) from the diary, for compliant and non-compliant participants.**

**Table 4. Analgesic consumption during the 14-day follow-up between participants receiving vitamin C and those receiving placebo for intention-to-treat and per-protocol analysis (perfect compliance).**

| Analgesic consumption | Vitamin C | Placebo |
|---|---|---|
| **Intention to treat analysis:** | N = 18 | N = 22 |
| Median (Q1-Q3) morphine 5mg pills consumed | 8.5 (3.8–20.0) | 6.0 (1.0–11.3) |
| *Diary information only* | N = 10 | N = 21 |
| Median (Q1-Q3) naproxen 500 mg pills consumed | 0 (0–3.3) | 15.0 (0–27.0) |
| Median (Q1-Q3) acetaminophen 650 mg pills consumed | 37.5 (0–68.8) | 28.0 (3.0–45.5) |
| **Per protocol analysis:** | N = 12 | N = 17 |
| Median (Q1-Q3) morphine 5mg pills consumed | 6.5 (3.3–19.5) | 9.0 (1.5–16.0) |
| *Diary information only* | N = 8 | N = 16 |
| Median (Q1-Q3) naproxen 500 mg pills consumed | 0 (0–9.8) | 20.0 (0–27.0) |
| Median (Q1-Q3) acetaminophen 650 mg pills consumed | 38.5 (5.0–76.3) | 34.5 (4.3–46.8) |

Q1-Q3: first and third quartiles; ED: emergency department; SD: standard deviation

## Daily pain intensity during follow-up

Mean pain intensity in each day for both groups and for intention-to-treat and per-protocol analyses are illustrated in Fig 3. For both types of analyses, the trends in pain intensity over the two weeks of the study were relatively similar for vitamin C and placebo groups.

## Discussion

Findings from this pilot study support the feasibility of conducting a full-scale randomized, placebo-controlled trial to determine the opioid-sparing and analgesic effects of vitamin C compared to placebo (lactose) over a two-week follow-up period for ED-discharged patients with acute musculoskeletal injuries. The recruitment rate was higher than expected. However, strategies to limit the high refusal and lost to follow-up rates should be planned for a future larger multicenter RCT.

The vitamin C and placebo groups were similar, except for a smaller percentage of females and fewer prescriptions of NSAIDs in the vitamin C group. Also, a higher percentage of acetaminophen was recommended in the vitamin C group. However, a small pilot study cannot be used to inform randomization performance [45]. Clinicians seemed to support the study since only 3 patients were missed out of 137 screened. The main barriers were the high refusal rate (41%) by eligible patients, mainly because they lacked interest for the study, were in too much pain and did not have enough time to participate. A recent review of RCTs reports a median (IQR) consent rate of 72% (50%–88%) which is much higher than our 59% consent rate [44]. Our results also show that patients with a shorter ED length of stay and those treated with an opioid during their ED stay seem to be more inclined to participate in the study. This aligns with the fact that patients are declining participation because they are in pain or have time constraints. Since patients are recruited when they are ED-discharged, providing effective analgesia and timely healthcare in the ED might help diminish the refusal rate in a future RCT. Also, to address the lack of interest in the study, providing more information on the waiting room monitors and leaflets with infographic visual aids could help.

Our median (IQR) retention rate of 75% is lower than the 88% (80%-97%) reported in the same review. Participants lost to follow-up appear to have a longer length of stay and are more often treated with opioids during their ED stay than participants who completed the study. Again, timely healthcare could limit this rate, but the higher rate of opioid treatment during ED visit in the lost to follow-up group is difficult to explain. This should be explored in a future

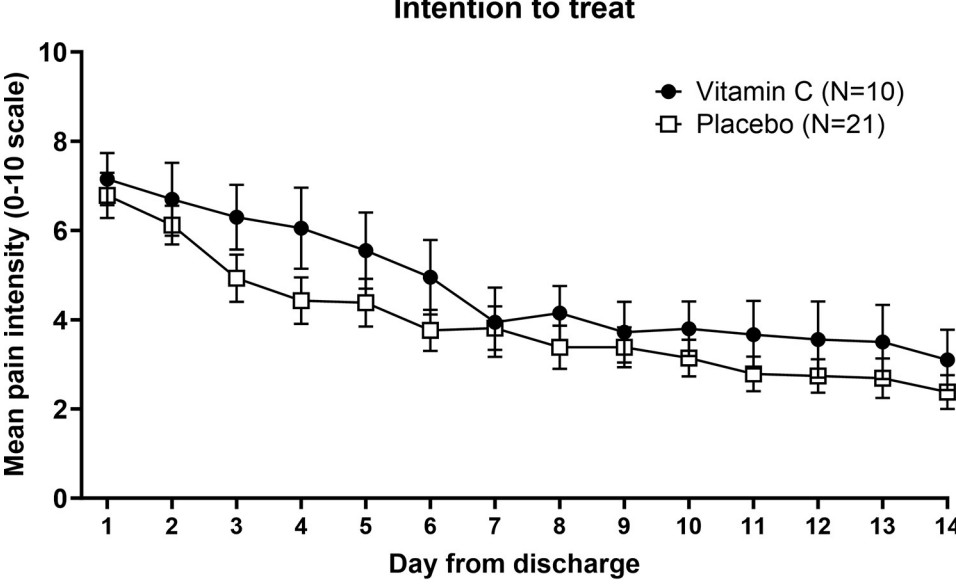

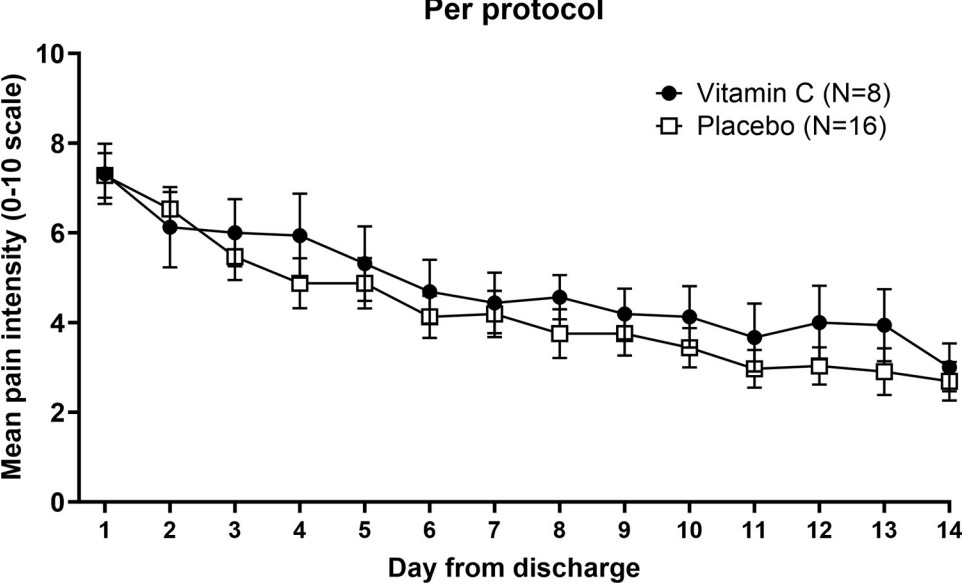

**Fig 3. Mean (SEM) daily pain intensity (0–10 scale) for intention to treat and per protocol analyses.**

RCT. Non-compliance with the consumption of study capsules was associated with less pain intensity in the second week of follow-up (Fig 2). It is reasonable to assume that patients experiencing little, or no pain may be less motivated to consume the study capsules identified as analgesics. Emphasizing the importance of consuming all study capsules, even if there is minimal or no pain, could help mitigate this issue. Finally, there appears to be an opioid-sparing effect of vitamin C in the per-protocol analysis. However, no conclusions should be drawn from this small sample.

## Limitations

This pilot RCT was not powered to detect differences in clinical outcomes, so any between-group comparison should be interpreted with caution. The wide confidence intervals and discordant results between intention-to-treat and per-protocol attest to this. The feasibility was evaluated only in one academic center, results could be different in non-academic or rural centers. In addition, we did not reach our planned sample size. This was due to an unexpectedly high refusal rate which depleted our funds. However, we have demonstrated the feasibility of a future large RCT and identified challenges to be addressed. Furthermore, the minimum sample size for a feasibility study is controversial and almost 40% of these studies do not achieve the sample size target [44].

## Conclusion

This pilot study supports the feasibility of a larger RCT on the opioid sparing and analgesic properties of vitamin C for acute MSK injured ED patients. However, strategies to reduce the refusal and lost to follow-up rate should be established.

## Supporting information

**S1 Table. Comparison of baseline characteristics between participants who were compliant and those who were not.**
(DOCX)

## Acknowledgments

We would like to thank Martin Marquis for English language editing.

## Author Contributions

**Conceptualization:** Raoul Daoust, David Williamson, Caroline Arbour, Jeffrey J. Perry, Marcel Émond, Judy Morris, Alexis Cournoyer.

**Formal analysis:** Jean Paquet, David Williamson.

**Funding acquisition:** Raoul Daoust, Jean Paquet, David Williamson, Vérilibe Huard, Caroline Arbour, Jeffrey J. Perry, Simon Berthelot, Alexis Cournoyer.

**Methodology:** Raoul Daoust, Jean Paquet, David Williamson, Judy Morris, Alexis Cournoyer.

**Project administration:** Raoul Daoust.

**Resources:** Simon Berthelot, Patrick Archambault, Dominique Rouleau.

**Supervision:** Raoul Daoust.

**Validation:** Vérilibe Huard.

**Writing – original draft:** Raoul Daoust, Jean Paquet.

**Writing – review & editing:** Raoul Daoust, Jean Paquet, David Williamson, Vérilibe Huard, Caroline Arbour, Jeffrey J. Perry, Marcel Émond, Simon Berthelot, Patrick Archambault, Dominique Rouleau, Judy Morris, Alexis Cournoyer.

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
