## [Decision Letter · Decision Letter 0]

16 Jul 2024

PONE-D-24-19288

Impact of Vitamin C on the Reduction of Opioid Consumption for Acute Musculoskeletal Pain: A Double-Blind Randomized Control Pilot Study

PLOS ONE

Dear Dr. Daoust,

Thank you for submitting your manuscript to PLOS ONE. After careful consideration, we feel that it has merit but does not fully meet PLOS ONE’s publication criteria as it currently stands. Therefore, we invite you to submit a revised version of the manuscript that addresses the points raised during the review process.

**ACADEMIC EDITOR: **

The authors conducted a double-blind randomized pilot study to evaluate the effect of vitamin C on reducing opioid consumption in patients treated in the emergency department for acute musculoskeletal pain. They compared a group receiving 1 g of vitamin C twice daily for 14 days to a placebo group, both with a standardized opioid prescription. The results show that while the median morphine consumption was similar between the two groups, the naproxen consumption was significantly lower in the vitamin C group. 

The manuscript is well-written, and the authors have appropriately acknowledged the limitations resulting from the small sample size.

1) I suggest authors to integrate the study rationale into the introduction section of the manuscript ; 

<ul><li> 

A rebuttal letter that responds to each point raised by the academic editor and reviewer(s). You should upload this letter as a separate file labeled 'Response to Reviewers'.<li> 

A marked-up copy of your manuscript that highlights changes made to the original version. You should upload this as a separate file labeled 'Revised Manuscript with Track Changes'.<li> 

We look forward to receiving your revised manuscript.

Kind regards,

Axel Benhamed, M.D, MSc

Academic Editor

PLOS ONE

 [This research was supported by the « Fonds Alma Mater et Chaire Docteur Sadok Besrour de l’Université de Montréal », the « Association des spécialistes en médecine d’urgence du Québec » and the « Fonds des Urgentistes de l’Hôpital du Sacré-Cœur de Montréal ».].  

3. In the online submission form, you indicated that [Data can be requested from authors permission.]. 

4. Please include your tables as part of your main manuscript and remove the individual files. Please note that supplementary tables (should remain/ be uploaded) as separate ""supporting information"" files".

Reviewers' comments:

Reviewer's Responses to Questions

**Comments to the Author**

1. Is the manuscript technically sound, and do the data support the conclusions?

Reviewer #1: Yes

Reviewer #2: Yes

Reviewer #3: Yes

Reviewer #4: Partly

2. Has the statistical analysis been performed appropriately and rigorously? 

Reviewer #1: Yes

Reviewer #2: Yes

Reviewer #3: Yes

Reviewer #4: I Don't Know

3. Have the authors made all data underlying the findings in their manuscript fully available?

Reviewer #1: Yes

Reviewer #2: Yes

Reviewer #3: Yes

Reviewer #4: Yes

4. Is the manuscript presented in an intelligible fashion and written in standard English?

Reviewer #1: Yes

Reviewer #2: Yes

Reviewer #3: Yes

Reviewer #4: Yes

5. Review Comments to the Author

Reviewer #1: As the statistical reviewer I will focus on methods and reporting. All looks reasonable but I was confused because the outcome for the future trial was not covered. one of the major aims of feasibility outcomes is to assess the outcome of interest for the future trial, obtain estimates of variability and reasonable effectiveness levels - that does not seem to be the case here or I have missed it. A section is needed on the next steps and what information could be used for the future trial, ideally a power calculation towards that.

Reviewer #2: General comments: This article presents a clinical trial evaluating the value of vitamin C in reducing morphine consumption in musculoskeletal pain. Patients are recruited for the study from the emergency departments in which they present. This article should be seen in the context of the "opioid crisis", which is affecting the USA in the main, but also Canada and to a lesser extent European countries.

The authors present this clinical trial as a feasibility study. It should make it possible to collect data on opioid and analgesic consumption in the study population after a visit to the emergency department for musculoskeletal trauma. It will also specify the pain scores of these patients over the fortnight following the emergency visit.

The study is monocentric. These quality criteria include prior publication of the protocol (reference 36) and adherence to CONSORT recommendations. Consecutive patients were recruited during the day on weekdays. There is no reason why this limitation of the inclusion period to the day should lead to a substantial bias in the results.

The authors' conclusions are correct for a feasibility study, identifying weaknesses in recruitment that need to be addressed. However, the authors should develop their hypothesis at greater length to explain the difference between per-protocol analysis and intention-to-treat analysis, which is theoretically the most appropriate method for analyzing a clinical trial. However, the intention-to-treat analysis, which shows a higher consumption of morphine in the vitamin C group, "in theory", does not really go in the direction intended by the authors. A lack of power cannot be invoked, since the inclusion of additional patients has methodologically no reason to change the direction of the difference. The authors therefore need to explain why they believe that the intention-to-treat analysis of this small feasibility study (which, it is true, is not designed to conclude on the efficacy of the treatment) is not predictive of a result in a future clinical trial.

Specific comments:

The authors' assertion associated with reference 25 that vitamin C is a drug with very low toxicity, even if globally acceptable, could be tempered by a recent article on vitamin C and sepsis in intensive care, which was associated in this clinical trial with excess mortality in the vitamin C group.

It seems to me that one piece of information is missing from the study's rationale: what is the percentage of opioid prescriptions in musculoskeletal trauma patients in the authors' department where this clinical trial is taking place, based on retrospective data, and what other therapeutic classes are used? These figures could be compared with, if available, similar data from other publications in Canada, the USA or Europe.

In the paragraph on vitamin C on page 6, the authors could indicate that the doses delivered are far in excess of physiological vitamin C requirements.

The authors indicate a median consent rate of 72% and a median retention rate of 88%. Could they clarify the definition of these two rates, as I don't understand why the retention rate is higher than the consent rate, unless the 88% is itself related to patients who have already consented.

In Table 2, the authors note several important differences between the vitamin C group and the placebo group (sex ratio, paracetamol, etc.). Would it be possible to have a column with the "p's" (as requested for the other tables)? If I've understood correctly, the authors attribute these differences to chance, due to the small number of patients compared?

The level of pain intensity above 7 confirms that this is severe pain, for which the recommendations are indeed to obtain a morphine prescription. However, pain intensity at admission cannot prejudge pain intensity at discharge from the emergency department, especially in patients with immobilization, which is itself highly analgesic. It would therefore be interesting to have the discharge VAS from Table 1.

Concerning pain intensity, as shown in figure 3, it can be seen that on average, both in per-protocol analysis and even more so in intention-to-treat analysis, pain was greater by between half and one VAS point each day in the vitamin C group. What was the statistical test used to compare these two series in figure 3? It seems to me that they can be treated as Kaplan-Meier curves.

Reviewer #3: Dear authors,

Thank you for the opportunity to review this article. The aim of this pilot study was to assess the feasibility of conducting a randomized, placebo-controlled trial to determine the opioid-sparing and analgesic effects of vitamin C compared to placebo(lactose) over a two-week follow-up period in ED-discharged patients with acute musculoskeletal (MSK) injuries.

Minor comments :

Some texts in the introduction seem to repeat themselves with slight nuances. For example, on page 3:

"Currently, the main strategy employed to reduce opioid consumption is to limit the rate and quantity of opioids prescribed for acute pain.[12-15] However, most interventions reduce opioid prescription rates, but not the quantity of prescribed opioids.[16]"

These two sentences should be combined to avoid repetition. Please review the flow of the introduction to eliminate this type of redundancy.

Methods :

* Patient and Public Involvement : please note the reference of ethic comitee.

Discussion :

Results should only appear in the results section. For example :

* "The main barriers were the high refusal rate (41%) by eligible patients, mainly because they lacked interest for the study, were in too much pain and did not have enough time to participate" OR "Our median (IQR) retention rate of 75% is lower than the 88% (80%-97%) reported in the same review"

Major comment :

To enhance the comprehensiveness of the article, I recommend incorporating a more detailed discussion of the pathophysiology, either in the introduction or the discussion section. This will provide a better understanding of the mechanisms by which vitamin C exerts its effects in this context.

Reviewer #4: I would like to thank the authors for giving me the opportunity to review this article.

The aim of this study was to assess the feasibility of conducting a randomized, placebo- controlled trial to determine the opioid-sparing and analgesic effects of vitamin C over a two-week follow-up period in ED-discharged patients with acute musculoskeletal injuries. So, the main outcome for feasibility was the recruitment rate. This work is very clearly presented and the article is well constructed. However, I fail to understand its purpose and the importance of the main result. Especially as the expected number of patients was not reached, with a high rate of loss to follow-up (25%), giving the opposite impression to that suggested by the authors. In this context, I find it very difficult to have an opinion on the scientific interest of such a publication.

6. PLOS authors have the option to publish the peer review history of their article (what does this mean?). If published, this will include your full peer review and any attached files.

Reviewer #1: No

Reviewer #2: **Yes: **JOLY Luc-Marie, Univ Rouen Normandy, Universitary Hospital of Rouen, Emergency Medicine Department, F-76000 Rouen, France

Reviewer #3: No

Reviewer #4: No

---

## [Author Response · Author response to Decision Letter 0]

23 Sep 2024

September 16, 2024

Revision for manuscript: ID: PONE-D-24-19288

Journal: PLOS ONE

Title: Impact of Vitamin C on the Reduction of Opioid Consumption for Acute Musculoskeletal Pain: A Double-Blind Randomized Control Pilot Study

Axel Benhamed, M.D, MSc

Academic Editor

PLOS ONE

Dear Dr Benhamed,

We would like to thank you for giving us the opportunity to resubmit a revised version of the manuscript. The co-authors and I have reworked the manuscript following the editor and reviewers’ recommendations. All comments and suggestions have been considered in our revisions and changes to the manuscript have been made using track changes. We also want to thank the reviewers for their suggestions, which helped us to significantly improve the paper’s quality.

Academic Editor

1) I suggest authors to integrate the study rationale into the introduction section of the manuscript; 

Response: Done. We removed the study rationale and study objectives headings in the introduction section. The study rational is now integrated into the introduction. 

Reviewer: 1

1) As the statistical reviewer I will focus on methods and reporting. All looks reasonable but I was confused because the outcome for the future trial was not covered. one of the major aims of feasibility outcomes is to assess the outcome of interest for the future trial, obtain estimates of variability and reasonable effectiveness levels - that does not seem to be the case here or I have missed it. A section is needed on the next steps and what information could be used for the future trial, ideally a power calculation towards that.

Response: Done. The outcome of interest for the future trial is the median number of morphine 5mg tablets consumed in the two groups and is reported in Table 4. In that table, we reported the median and the 25-75 quartiles for each group. We now added the effect size related to the treatment effect in the results section even if the study was not designed to conclude on the efficacy of the treatment. “For the intention to treat analysis, the median number of morphine 5mg tablets consumed was higher in the vitamin C group than the placebo group (8.5 vs 6.0 tablets; effect size: r = 0.27). and when we examine the per-protocol analysis, the median number of morphine 5mg tablets consumed was lower in the vitamin C group than the placebo group (6.5 vs 9.0 tablets; effect size: r = -0.11).”

We also added in the statistical analysis section a sentence to explain how we calculate the effect size from median differences: “We calculated the treatment effect size by using the formula r = z/√N, where z comes from the Mann-Whitney U test.”

Reviewer: 2

However, the authors should develop their hypothesis at greater length to explain the difference between per-protocol analysis and intention-to-treat analysis, which is theoretically the most appropriate method for analyzing a clinical trial. However, the intention-to-treat analysis, which shows a higher consumption of morphine in the vitamin C group, "in theory", does not really go in the direction intended by the authors. A lack of power cannot be invoked, since the inclusion of additional patients has methodologically no reason to change the direction of the difference. The authors therefore need to explain why they believe that the intention-to-treat analysis of this small feasibility study (which, it is true, is not designed to conclude on the efficacy of the treatment) is not predictive of a result in a future clinical trial. 

Response: We did address the difference in intention-to-treat and per-protocol results in the limitations section: “This pilot RCT was not powered to detect differences in clinical outcomes, so any between-group comparison should be interpreted with caution. The wide confidence intervals and discordant results between intention-to-treat and per-protocol attest to this. We also added that “Since non-compliance is associated with less pain, particularly in the second week of treatment, this could partly explain the discordant results observed in our small sample.”

Specific comments:

The authors' assertion associated with reference 25 that vitamin C is a drug with very low toxicity, even if globally acceptable, could be tempered by a recent article on vitamin C and sepsis in intensive care, which was associated in this clinical trial with excess mortality in the vitamin C group.

Response: The LOVIT study was done in a very different population and much higher dosage of vitamin C “proven or suspected infection as the main diagnosis, and who were receiving a vasopressor to receive an infusion of either vitamin C (at a dose of 50 mg per kilogram of body weight) … every 6 hours”. This excess of mortality as not been reported at dosage used in our study.

It seems to me that one piece of information is missing from the study's rationale: what is the percentage of opioid prescriptions in musculoskeletal trauma patients in the authors' department where this clinical trial is taking place, based on retrospective data, and what other therapeutic classes are used? These figures could be compared with, if available, similar data from other publications in Canada, the USA or Europe.

Response: We agree that this information could be of interest to the reader. However, the information on physician prescription after ED discharge is not enter on our database system. Therefore, that information is not available.

In the paragraph on vitamin C on page 6, the authors could indicate that the doses delivered are far in excess of physiological vitamin C requirements.

Response: Done. We added this in the paragraph on vitamin C: “However, this dosage exceeds the adult Recommended Dietary Allowances set by the Government of Canada, which is 65 to 90 milligrams per day.” 

The authors indicate a median consent rate of 72% and a median retention rate of 88%. Could they clarify the definition of these two rates, as I don't understand why the retention rate is higher than the consent rate, unless the 88% is itself related to patients who have already consented.

Response: The 88% relates only to the patients who consented. To clarify this, we changed the phrase “To assess feasibility, our results were compared to those reported in a recent review of randomized controlled trials; 72% (IQR: 50%–88%) median consent rate of eligible patients and 88% (IQR: 80%–97%) median retention rate of patients who consented.”

In Table 2, the authors note several important differences between the vitamin C group and the placebo group (sex ratio, paracetamol, etc.). Would it be possible to have a column with the "p's" (as requested for the other tables)? If I've understood correctly, the authors attribute these differences to chance, due to the small number of patients compared?

Response: The information in Table 2 and in the other tables is for descriptive purposes only. They are not statistically tested because they are not powered and they are not subject to hypothesis testing.

The level of pain intensity above 7 confirms that this is severe pain, for which the recommendations are indeed to obtain a morphine prescription. However, pain intensity at admission cannot prejudge pain intensity at discharge from the emergency department, especially in patients with immobilization, which is itself highly analgesic. It would therefore be interesting to have the discharge VAS from Table 1.

Response: The information in Table 1 compares patients who refused to participate with those who participated in our study. Pain intensity at discharge for patients who refused to participate is not available.

Concerning pain intensity, as shown in figure 3, it can be seen that on average, both in per-protocol analysis and even more so in intention-to-treat analysis, pain was greater by between half and one VAS point each day in the vitamin C group. What was the statistical test used to compare these two series in figure 3? It seems to me that they can be treated as Kaplan-Meier curves.

Response: For our main outcome shown in Table 4, pain intensity over the 14 days was reported for descriptive purposes only. In a pilot study, treatment effects are not statistically tested because of power issues and because it is not intended to test treatment effect. We report only the mean and SEM for each group to aid in the design of a future multicenter randomized trial.

Reviewer: 3

Some texts in the introduction seem to repeat themselves with slight nuances. For example, on page 3: "Currently, the main strategy employed to reduce opioid consumption is to limit the rate and quantity of opioids prescribed for acute pain. [12-15] However, most interventions reduce opioid prescription rates, but not the quantity of prescribed opioids. [16]"

Response: These sentences do not communicate the same thing. The first describes approaches and the second describes the effects of those approaches.

Patient and Public Involvement: please note the reference of ethic comitee.

Response: Done. We added the name of the ethic committee and the study reference number: “This study was approved by the local Ethics Review Committee (Comité d'éthique de la recherche du CIUSSS du Nord-de-l'Île-de-Montréal: MP-32-2023-2442).” 

Results should only appear in the results section. For example :

"The main barriers were the high refusal rate (41%) by eligible patients, mainly because they lacked interest for the study, were in too much pain and did not have enough time to participate" OR "Our median (IQR) retention rate of 75% is lower than the 88% (80%-97%) reported in the same review"

Response: Done. Those numbers are now removed from the discussion section. 

To enhance the comprehensiveness of the article, I recommend incorporating a more detailed discussion of the pathophysiology, either in the introduction or the discussion section. This will provide a better understanding of the mechanisms by which vitamin C exerts its effects in this context.

Response: Done. In the introduction section we added “There is no consensus as to its analgesic mechanism. The most widely accepted mechanism involves reducing free radicals’ production, which helps protect tissues (including nerves) from irreversible damage. It may also be a cofactor for the biosynthesis of opioid peptides, and modulate the expression of genes involved in pain perception.”

Reviewer: 4

So, the main outcome for feasibility was the recruitment rate. This work is very clearly presented and the article is well constructed. However, I fail to understand its purpose and the importance of the main result. Especially as the expected number of patients was not reached, with a high rate of loss to follow-up (25%), giving the opposite impression to that suggested by the authors. In this context, I find it very difficult to have an opinion on the scientific interest of such a publication.

Response: The main purpose of this type of study is to inform the design and risk of failure/success of future larger similar trials. If a researcher reads a feasibility study that shows that a full trial on a particular topic would be unlikely to meet recruitment/retention rates and other feasibility outcomes, then the potential wasted cost of that future trial is avoided, or the planned trial design can be modified to avoid the obstacles identified in the feasibility study. On the other hand, if the feasibility study shows that a full trial is likely to be successfully completed, researchers planning similar studies can be assured that the cost and effort are likely to be worthwhile.

We thank you for the review of this manuscript. We hope that our revised version meets your standards, and that the manuscript is now suitable for publication. 

Please do not hesitate to contact the undersigned for further information.

Best regards,

Raoul Daoust, MD, MSc 

Corresponding Author 

Emergency Department

Sacre-Cœur Hospital

5400 Gouin blvd. West

Montreal (Qc) Canada H4J 1C5

E-mail: raoul.daoust@videotron.ca

---

## [Decision Letter · Decision Letter 1]

11 Dec 2024

Impact of Vitamin C on the Reduction of Opioid Consumption for Acute Musculoskeletal Pain: A Double-Blind Randomized Control Pilot Study

PONE-D-24-19288R1

Dear Dr. Raoul Daoust

We’re pleased to inform you that your manuscript has been judged scientifically suitable for publication and will be formally accepted for publication once it meets all outstanding technical requirements.

Kind regards,

Axel Benhamed, M.D, MSc

Academic Editor

PLOS ONE

Additional Editor Comments (optional):

Dear Authors,

Thank you for submitting the revised version of your manuscript to PLOS ONE and for the effort you have dedicated to addressing the reviewers’ comments.

After a thorough review of your revisions, I am pleased to note that you have adequately and thoughtfully addressed the concerns raised during the initial assessment. Your responses are well-reasoned, detailed, and enhance the overall quality and clarity of the manuscript.

I am therefore delighted to inform you that your article has been accepted for publication in PLOS ONE.

Congratulations on your work, and thank you for choosing PLOS ONE as the platform to share your research. We look forward to seeing the impact of your findings within the scientific community.

Best regards,

Reviewers' comments:

Reviewer's Responses to Questions

**Comments to the Author**

1. If the authors have adequately addressed your comments raised in a previous round of review and you feel that this manuscript is now acceptable for publication, you may indicate that here to bypass the “Comments to the Author” section, enter your conflict of interest statement in the “Confidential to Editor” section, and submit your "Accept" recommendation.

Reviewer #1: All comments have been addressed

Reviewer #5: All comments have been addressed

2. Is the manuscript technically sound, and do the data support the conclusions?

Reviewer #1: Yes

Reviewer #5: Yes

3. Has the statistical analysis been performed appropriately and rigorously? 

Reviewer #1: Yes

Reviewer #5: I Don't Know

4. Have the authors made all data underlying the findings in their manuscript fully available?

Reviewer #1: (No Response)

Reviewer #5: Yes

5. Is the manuscript presented in an intelligible fashion and written in standard English?

Reviewer #1: Yes

Reviewer #5: Yes

6. Review Comments to the Author

Reviewer #1: I am satisfied with the authors' responses and the resulting changes to the paper.......................

Reviewer #5: Thanks for the opportunity to review this revised manuscript. I have read through the comments from previous reviewers and the revised manuscript and agree that all comments have been addressed. I am not as concerned as the previous reviewers about the clinical outcomes being reported as per protocol versus intention-to-treat analysis, as this is not the primary concern of a pilot trial. I would encourage the authors to report the main trial as intention-to-treat, unless they have a very well justified reason to use per-protocol. I look forward to seeing further investigation on this interesting topic.

Best wishes,

Caitlin

7. PLOS authors have the option to publish the peer review history of their article (what does this mean?). If published, this will include your full peer review and any attached files.

Reviewer #1: No

Reviewer #5: **Yes: **Caitlin Jones

---

## [Editor Report · Acceptance letter]

16 Dec 2024

PONE-D-24-19288R1 

PLOS ONE

Dear Dr. Daoust, 

I'm pleased to inform you that your manuscript has been deemed suitable for publication in PLOS ONE. Congratulations! Your manuscript is now being handed over to our production team.

Kind regards, 

on behalf of

Dr. Axel Benhamed 

Academic Editor

PLOS ONE